# Impact of a DANA Event on the Thermal Response of Nectarine Trees

**DOI:** 10.3390/plants12040907

**Published:** 2023-02-17

**Authors:** María R. Conesa, Wenceslao Conejero, Juan Vera, Ana Belén Mira-García, María Carmen Ruiz-Sánchez

**Affiliations:** Irrigation Department, CEBAS-CSIC, Campus de Espinardo 25, 30100 Murcia, Spain

**Keywords:** canopy temperature, DANA, leaf–water relations, soil water content, flooding, nectarine orchard

## Abstract

This field experiment focuses on the effects of a heavy rainfall event (DANA, *depresión aislada en niveles altos)* that occurred on 12–14 September 2019 (DOY, Day of the year, 255–257), in southern Spain on plant water status and the thermal response of nectarine trees. Two irrigation treatments were applied during the summer–autumn postharvest period (DOY 158–329): full-irrigated (CTL) and non-irrigated (DRY). Volumetric soil water content (θ_v_), air temperature (Ta) and canopy temperature (Tc) were monitored in real-time and the crop water stress index (CWSI) was calculated. The difference in Tc between the DRY and CTL treatments (Tc’ − Tc) is proposed as a new thermal indicator. Stem water potential (Ψ_stem_) and leaf gas exchange measurements were recorded on representative days. During the DANA event, only the Tc measured by the infrared radiometer sensors could be monitored. Therefore, the effects of the DANA forced the soil water content sensors to be switched off, which prevented Ψ_stem_ and leaf gas exchange determinations from DOY 255 to 275. Before the DANA event, withholding irrigation caused a gradual decrease in the soil and plant water status in the DRY treatment. Significant differences appeared between treatments in the studied thermal indexes. Moreover, Tc’ − Tc was more sensitive than Tc − Ta in assessing nectarine water stress. The effects of the DANA reduced these differences, suggesting different baselines for the calculation of CWSI. In this respect, the relationship Tc − Ta vs. VPD improved the coefficient of determination after the DANA event in full-irrigated trees. Similar values of Ψ_stem_ and leaf gas exchange were found in both treatments after the DANA event, even though thermal indexes showed some significant differences. In addition, the strong relationship found between Tc − Ta and CWSI vs. Ψ_stem_ worsened after DANA occurred, revealing a lower sensitivity of Ψ_stem_ compared to canopy temperature to accurately assess nectarine water status in these saturated soil conditions. This research underlined the robustness of infrared thermography to continuously monitor plant water status under these extreme weather conditions.

## 1. Introduction

Extreme events associated with climate change are now being experienced with greater frequency and intensity worldwide, especially in the countries of the Mediterranean basin [1]. Heavy rains cause widespread flooding and very serious damage affecting the entire population. In recent years, DANA events (Spanish acronym for *depresión aislada en niveles altos*, meaning upper-level isolated atmospheric depression) have replaced the traditional torrential rains and flash-flooding associated with cold drops [2]. Technically, it is a depression isolated at high levels of the atmosphere, where a cold front clashes with warm air, causing torrential rainfalls. In practice, a DANA event generates high-intensity storms causing violent run-offs with a high capacity for soil erosion and flooding. Moreover, it has been shown that the effects of a DANA event are much more devastating than those of a cold drop [3].

On the 12–14 September 2019, a DANA event struck the Southeast of Spain, affecting the Region of Murcia (the study area of this work) very intensely. More than 400 mm was recorded in just 24 h, which exceeds the annual average of the historical series [4]. Moreover, being a semi-arid region where vegetation cover is not abundant, the extreme rainfall promoted significant soil erosion [4], aggravating the damage caused by the torrential rainfalls. A pronounced increase in the percentage of precipitation amount due to heavy (>30 mm day^−1^) and extreme (>50 mm day^−1^) precipitation was also observed during the last decade [1]. Paxian et al. [5], using regional climate models extending up to 2050, showed that in the autumn–winter there is a distinct tendency towards more intense precipitation extremes in many northern Mediterranean regions, particularly over Spain and Turkey. Thus, regions with traditional reduced precipitation amounts will be confronted with heavier individual events. 

Remote sensing techniques have been used to study and monitor natural weather disasters [6]. Scientific research in the field of remote sensing of heavy rainfall and floods has increased considerably in recent years, as innovative methods and equipment have been developed to record real-time information from multi-scale coverages [7]. Due to the limitations caused by floods, thermal sensors compared to soil-based ones (e.g., soil water content, matric potential sensors or flowmeters), are very valuable for monitoring the impact of this extreme weather event on crops [6]. 

At the field-scale in agriculture, infrared canopy thermography has emerged as a promising technology to continuously monitor plant water status in a non-destructive and cost-effective way [8]. This technique is based on the leaf energy balance by measuring canopy temperature (Tc). When a water deficit situation is applied, plants respond with a partial stomatal closure, reducing the stomatal conductance (g_s_), limiting leaf transpiration and promoting an attenuation of the evaporative cooling process, resulting in higher Tc values [9]. The main advantage of using Tc compared to other traditional plant-based water status indicators such as midday stem water potential (Ψ_stem_) or g_s_, is its real-time monitoring capacity together with its reliability [10]. However, Tc does not only depend on the stomatal aperture but also on other agrometeorological variables such as solar incidence angle, solar radiation, air temperature (Ta) and wind speed [11]. For this reason, the canopy-to-air temperature difference (Tc − Ta, Idso et al. [12]), and the crop water stress index (CWSI, Jackson et al. [13]) have been developed to minimize the effects of environmental variables on the absolute Tc values. CWSI is inversely related to the transpiration rate and g_s_ [14], but has been successfully related to plant-based water status indicators. Different relationships have been established to predict plant water status based on the Tc or CWSI values in fruit trees such as almond [15,16], peach and nectarine [17,18], apple [19], pistachio [20], sweet cherry [8] and papaya [21], among others. Nevertheless, in fruit trees, stomatal control of the canopy conductance and plant water status is highly sensitive to the vapor pressure deficit (VPD), as described in Conesa et al. [22], varying throughout crop development and between cultivars [18,23]. Subsequently, CWSI requires the use of two baselines that relate the canopy temperature under maximum stress and non-water stress conditions with the vapor pressure deficit (VPD), using data on clear sunny days and in full-irrigated conditions [10,13]. 

To our knowledge, no studies have reported the effects of DANA events on the thermal response of fruit trees. This work addresses the effects of a DANA event, which occurred on the 12–14 September 2019, on the physiological and thermal response of field-grown nectarine trees that had been subjected to full-irrigated and drought conditions. Given that this natural disaster modifies the physiological response of trees, we hypothesized that the non-water stressed baseline for CWSI calculations should be different before and after this extreme weather event. Furthermore, the use of infrared thermography to monitor plant water status under this type of natural weather disaster is examined with special attention.

## 2. Results and Discussion

### 2.1. Environmental Conditions and Soil Water Content 

As usual in the Mediterranean environment, the meteorological conditions during the postharvest period of early maturing cultivars (May–October) were more demanding in summer than in autumn, with a mean air temperature (Ta) of 26 °C in the summer and 18 °C in autumn. Maximum values of Ta (31.1 °C), vapor pressure deficit (VPD = 3.55 kPa) and reference crop evapotranspiration (ET_0_ = 7.67 mm day^−1^) were reached on DOY 221. The minimum values were recorded in autumn (DOY 326), with mean values of Ta = 11 °C, VPD = 0.43 kPa and ET_0_ = 0.05 mm day^−1^ (Figure 1A). More importantly, a total of 531.2 mm of rainfall was recorded during the experiment. In fact, 470.4 mm of rainfall were recorded between DOY 255 and 256 because of the DANA phenomenon. This event occurs normally in the autumn due to convective storms generated by the existence of cold air in the upper layers of the atmosphere combined with warm winds from the Mediterranean Sea [2]. In addition, another heavy rainfall event (19 mm) occurred on DOY 295. 

Considering that irrigation in the CTL treatment was suspended from DOY 255 to 275 due to the flooding caused by the DANA event, a total irrigation volume of 473 mm was applied to the CTL treatment, whereas the irrigation in the DRY treatment was withheld from June to November. Vera et al. [24] reported that the annual water requirements of early-maturing nectarine trees in the agro-climatic Region of Murcia (SE Spain) were around 660 mm, with the irrigation frequency varying from 1 to 7 days per week depending on the phenological period. 

As shown in Figure 1B, the real-time monitored volumetric soil water content (θ_v_) values in the CTL treatment were close to the field capacity, ranging from around 27–32%, while the irrigation withholding in the DRY treatment caused a gradual decrease in θ_v_, reaching a minimum value of 13% in DOY 246, close to the wilting point. When the DANA occurred, the capacitance probes were disconnected to avoid any short circuit in the telemetry system. Thus, no θ_v_ data was recorded from DOY 255 to 275. After that, θ_v_ values in the CTL treatment were above the field capacity until the end of the experiment. Meanwhile, the DRY treatment showed a θ_v_ value of 18.3% when reconnected and even decreased after that. Due to the rainfall recorded in DOY 295, θ_v_ levels in the DRY treatment significantly increased, followed by a subsequent decrease until the end of the experiment. The θ_v_ data also revealed that the soil water content varied not only because of irrigation or rainfall, but also fluctuated following diurnal environmental changes and root water uptake dynamics. In fact, the θ_v_ dynamics had been closely related to evapotranspiration demand, confirming the sensitivity of this type of capacitance sensor to the close environment of soil and plant roots [25,26].

It is also observed that irrigation in the CTL treatment was reestablished 20 days after the DANA event, coinciding with the rehabilitation of soil water sensors in both treatments.

### 2.2. Plant Water Status and Leaf Gas Exchange

Plant water status was estimated by the midday stem water potential (Ψ_stem_) (Figure 2A). Prior to the DANA event, the CTL treatment showed an average Ψ_stem_ value of about −0.80 MPa, which was indicative of non-limiting soil water conditions, as has been widely described for peach and nectarine trees [22,23,24,25,26,27,28,29,30]. The lowest Ψ_stem_ values were measured during the summer in response to the increased evaporative demand, when the maximum VPD values were recorded (Figure 1A and Figure 2A). Therefore, a strong decrease in the Ψ_stem_ was observed in the DRY treatment due to irrigation withholding, with an average of −2.08 MPa in this period. In this sense, the greatest plant water deficit (−2.92 MPa) was observed in DOY 199. Differences with respect to the CLT treatment were constant and statistically significantly different from DOY 169 (second day of measurement) onwards. The maximum Ψ_stem_ difference between the two treatments was 1.80 MPa (DOY 246). Additionally, the DANA event prevented discrete plant measurements until DOY 275. When the flooding ceased and the field determinations were returned, the Ψ_stem_ values in the DRY treatment reached levels of the full-irrigated trees, averaging −0.70 MPa in this period (Figure 2A). In this regard, the threshold values of Ψ_stem_ of −1.5 and −2.0 MPa during the postharvest are recommended to ensure no impairment of bloom fertility [31] and to limit the occurrence of double fruits [32], respectively. 

Before the occurrence of the DANA event, seasonal variations in leaf gas exchange were more pronounced than those observed for Ψ_stem_, with mean values for net photosynthesis (P_n_) and stomatal conductance (g_s_) of ≈ 17.5 µmol m^−2^ s ^−1^ and 273.5 mmol m^−2^ s ^−1^ for the CTL treatment, respectively (Figure 2A,B). These values agreed with those reported in early-maturing peach and nectarine trees under non-limiting soil water conditions by other studies [22,23,24,25,26,27,28,29,30]. In the DRY treatment, the P_n_ and g_s_ values were around 28% and 36% lower than in the CTL treatment, respectively. Rahmati et al. [28] observed a reduction (>50%) in the leaf gas exchange when Ψ_stem_ decreased from −1.4 to −2.0 MPa in peach trees. However, an additional Ψ_stem_ decrease below −2.0 MPa led to only a slight decrease in both the g_s_ and P_n_. Shackel et al. [33] observed a Ψ_stem_ threshold value of −1.5 MPa associated with a decrease in P_n_ that was compensated by a reduction in the vegetative apex growth. In our study, flash-flooding derived from the DANA event restored the leaf gas exchange in the DRY treatment to levels of the CTL treatment. However, in the accompanying research [22], the withholding irrigation treatment did not show a recovery in the values of g_s_ and P_n_ after the restored irrigation. 

As expected, the intrinsic and instantaneous water use efficiencies (P_n_/g_s_ and P_n_/transpiration, E) increased in the DRY treatment (Figure 2E,F). The maximum values of P_n_/g_s_ (137.9 μmol mol^−1^) and P_n_/E (3.91 μmol mmol^−1^) were obtained before DANA, on DOY 246. Most woody species increase their water use efficiency as an additional mechanism of drought acclimation [34]. These results indicated that the water loss by transpiration during the water deficit was a key regulator of the drought response [35], suggesting a predominant stomatal control over photosynthesis [36]. Although we do not have leaf gas exchange data when the DANA event occurred, a decrease in P_n_ and g_s_ would have been expected due to low root temperatures. According to Delucia [37], the main factor of this reduction is leaf water limitation resulting in an increase in the viscosity of water and in the root resistance. Additionally, in the same nectarine trees, Conesa et al. [22] found that P_n_/g_s_ was more sensitive than P_n_/E as a plant water status indicator in detecting water deficit situations. Furthermore, the study revealed that E, apart from stomatal closure, is also driven by other factors such as boundary layer conductance, which is mediated by VPD through changes in the wind speed. 

### 2.3. Canopy Temperature and Thermal Indexes

Real-time canopy temperature (Tc) values are shown in Figure 3. During the postharvest period, continuous Tc records in nectarine trees varied from 14 to 47 °C in the summer and from 4 to 35 °C in autumn. Minimum Tc values were recorded at the beginning and end of the day, whereas maximum Tc values were recorded around midday. Before the DANA, the Tc values in CTL and DRY treatments at midday averaged at 38.6 °C and 44.1 °C, respectively. A mean difference of 3 °C was registered in this period between the two treatments. However, the effects of the DANA minimized these differences, restoring Tc in the DRY treatment to the levels of full-irrigated trees (≈27 °C). After the DANA, a progressive increase in the Tc values of the DRY treatment would have been expected as the soil dried out, but this did not happen, perhaps due to the progressive decrease in the evaporative demand and Ta observed during this period (Figure 1A). 

At the present time, the canopy temperature can be measured on the ground (surface-scale) and using remote sensing technologies. On the ground, Tc is usually performed with infrared radiometer sensors (as in our case), hand-held thermal guns and thermography cameras, while thermal imagery by remote sensing is performed with unmanned aerial vehicles (UAVs) and satellites [38]. As reported by Ramirez-Cuesta et al. [30], one of the main problems of using ground-based indicators is the huge number of measurements needed to completely cover the study area to characterize the spatial variability of plant water status among trees in an orchard and also within a single tree canopy. This variability depends on many factors, including the physical–chemical properties of soil and plant morphology (canopy size and architecture), among others [39]. Although this statement is correct, the use of UAVs equipped with hyperspectral cameras would not have been possible to cope with a DANA event, such as the one described in this study. They require pre-flight planning as well as good weather conditions. In fact, the main limitation of airborne thermography refers to the instantaneous nature of thermal sensors, which provides measurements at a specific moment [8,30]. For example, in the case of satellite images, there must be a coincidence between the time of the field measurement and the exact time at which the satellite orbits over the area, which is unlikely, especially when the satellite advances or delays its time of passage as the days pass [38]. Moreover, the fact that infrared radiometer sensors are able to continuously monitor Tc differentiates them from other on-ground thermal devices. Recently, Giménez-Gallego et al. [40] have developed a new sensor capable of automatically measuring Tc using a low-cost thermal camera, but its use in field conditions is still beginning. 

Since Tc is influenced by atmospheric conditions such as the vapor pressure deficit and radiation level among others [12], the use of Tc as a plant water status indicator requires normalization to account for varying environmental conditions [13]. To address this, the canopy-to-air temperature difference (Tc − Ta) can be used as an indicator of nectarine water status [18]. Furthermore, in this work, we proposed the canopy temperature difference between DRY and CTL treatments (Tc’ − Tc) as a new thermal indicator to mitigate the effect of environmental conditions on the absolute values of Tc (Figure 4). 

Both thermal indicators tended to decrease as the experiment progressed due to the less demanding environmental conditions and the degree of tree defoliation, which was also influenced by the torrential rainfalls. As expected, the largest daily differences in Tc − Ta between treatments were obtained before the DANA event. In this period, the maximum Tc − Ta observed was 0.8 °C and 3.8 °C in the CTL and DRY treatments, respectively. The maximum Tc − Ta values that occurred at midday were also reported by Massai et al. [41]. Wang and Gartung [17] measured Tc − Ta values in the 5–7 °C range and 1.4–2 °C in stressed and control peach trees, respectively. The negative Tc − Ta values observed in the CTL treatment indicate that an evaporative cooling process was taking place. Heat is dissipated by the water loss from the stomata via transpiration, but as soon as the stomata close, the canopy temperature raises several degrees above the air temperature [10,19]. In addition, the maximum value of Tc’ − Tc was 5.9 °C (Figure 4). The simplicity of this indicator could favor its use as a preliminary indicator of the water deficit. After the DANA event, the significant differences between the CTL and DRY treatments were reduced due to the recovery of Tc − Ta and Tc’ − Tc values in the DRY treatment. Interestingly, the values of Tc − Ta in the DRY treatment were close to 0, which is indicative of full-irrigated trees. This agreed with the plant water status and the performance of the leaf gas exchange, which were indicative of the absence of water deficit from the onset of the DANA event to the end of the experiment (Figure 2). The recognition of Tc − Ta as the most friendly thermal indicator to ascertain the plant water status and to monitor the physiological behavior of fruit trees has been reported in almond [15,16], peach and nectarine [17,18], apple [19], pistachio [20], sweet cherry [8] and papaya [21]. In this work, we suggest Tc’ − Tc as a new thermal indicator to monitor the nectarine water status. 

In line with these findings, a recent work by Ramírez-Cuesta et al. [30] compared the feasibility of ground- and airborne-based thermal indicators to identify and quantify the water status of a peach orchard under different irrigation regimes. The authors concluded that Ψ_stem_ was a better indicator of the plant water status using remote sensing than from ground measurements, because of the reduction in the spatial variability, but ground measurements better explained the changes in leaf gas exchange than airborne thermal data due to the higher spatial resolution of the proximal ground thermal sensor. During the experiment, our thermal data from infrared radiometer sensors yielded similar results for Ψ_stem_ and leaf gas exchange in both treatments, and were also able to continuously monitor Tc during the DANA. 

Table 1 includes the linear relationships between Tc − Ta vs. VPD, and Tc’ − Tc vs. VPD at different intervals of time during the day. In general, the results were highly time-dependent, as observed by Egea et al. [42]. For the CTL nectarine trees, results have shown that the best correlation was in the interval of 08:00–10:00 h GTM + 0, before the DANA event (R^2^ = 0.72 ***), and it improved after the DANA event (R^2^ = 0.84 ***). From this time, the linear regressions worsened considerably before the DANA, while they remained in the same range of R^2^ until the early afternoon (16:00–18:00 h GTM + 0) during the post-DANA period. Interestingly, the proposed thermal indicator of Tc’ − Tc improved the relationships with VPD during all time slots, compared to those obtained with Tc − Ta. Anyway, the best adjustment was also observed at 08:00–10:00 h GTM + 0. This agrees with the stomatal sensitivity of the *Prunus* species, showing their highest values of P_n_ and g_s_ in the early morning, coupled with higher light saturation and moderate Ta values [43]. From data in the interval 08:00–10:00 h GTM + 0, Figure 5 represents the lower limit (LL) taken from the Tc − Ta vs. VPD relationship (named Non-Water Stressed Baseline, NWSB) of CTL nectarine trees. To obtain the upper limit (UL), the Tc − Ta vs. VPD regressions for the DRY treatment were depicted. This revealed quasi-parallel lines to the VPD axis, with very low slopes in both situations (before and after the DANA). Meanwhile, the UL increased the fit obtained after the DANA. The upper limit could be assumed from the intercept of the NWSB corrected for Ta [10,44] or as the maximum Tc − Ta observed of a known highly stressed canopy [45]. Moreover, some authors have used a constant as the UL, assuming that the canopy is always above the air temperature when leaf transpiration is suspended, for instance Ta + 4 °C in grapefruit trees [46] or Ta + 5 °C in olive trees [47]. These results demonstrate the importance of considering a different NWSB when a natural weather disaster, such as a DANA phenomenon, occurs. In addition, robust upper and lower limits are needed to accurately characterize the CWSI with precision [48].

The seasonal pattern of the crop water stress index (CWSI) showed a trend similar to that of the canopy temperature readings (Figure 6). The LL was taken from the Tc − Ta vs. VPD relationship of full-irrigated nectarine trees. These limits help to normalize Tc to microclimatic conditions and can greatly influence the accuracy of Tc for stress detection [49]. The 15-min data suggested a hysteresis phenomenon, with a higher Tc − Ta in the morning (denoting warming) and a lower one in the afternoon (denoting cooling). The UL was selected for the same time interval as that of the LL of stressed nectarine trees. The maximum difference of the CWSI between the two treatments was observed before the DANA event. Thus, the highest CWSI value (0.94) in the DRY treatment was observed on DOY 232, while the CTL treatment averaged a CWSI value of 0.31. As with the other thermal indicators, the DANA event reduced the existing differences in CWSI between treatments. In this sense, the CWSI showed average values of 0.15 and 0.30 for CTL and DRY treatments during the post-DANA period. A CWSI close to 0 is indicative of a minimal stress level. Compared to Tc − Ta, the performance of the CWSI is improved by the consideration of the evaporative demand, as well as the species-specific response of water relations to VPD [44]. García-Tejero et al. [15] reported that the CWSI is more robust than Tc − Ta, especially under more variable environmental conditions throughout the day.

The thermal indexes ΣTc − Ta and ΣTc’ − Tc integrated Tc − Ta and Tc’ − Tc values on a daylight time basis (Figure 7). They gradually increased during the experiment and were higher in Tc’ − Tc > Σ_(DRY)_, Tc − Ta > Σ_(CTL),_ Tc − Ta, respectively. Since the CTL treatment was fully irrigated, the seasonal trend of ΣTc − Ta practically did not vary during the experiment. Meanwhile, the ΣTc’ − Tc was 40% higher than ΣTc − Ta in the DRY treatment, revealing the sensitivity of this thermal indicator to detect water deficit situations. The fact that higher ΣTc’ − Tc values were associated with a higher plant water stress (more negative Ψ_stem_ values) reinforces the continuous Tc measurements based on infrared sensors as a good water status indicator in fruit trees. Given that Tc’ − Tc is more sensitive to detect stress conditions than Tc − Ta, our findings showed that Tc’ − Tc can be a suitable plant-based water status indicator for irrigation management in Mediterranean areas with water scarcity. 

### 2.4. Thermal Indexes vs. Plant Water Status

Different linear regression models were calculated between the thermal indicators (Tc, Tc − Ta and CWSI) and Ψ_stem_ prior to and after the DANA occurred (Figure 8, Figure 9 and Figure 10). The relationship Tc’ − Tc vs. Ψ_stem_ was not practicable. Before DANA, two different clouds of points, corresponding to CTL and DRY treatments, were observed. Thus, higher Tc values were related to stressed nectarine trees. Although changes in Tc were explained by 48% of the plant water status, Tc showed some limitations as a water stress indicator. It is known that the water deficit in fruit trees induces stomatal closure and increases canopy temperature [8]. However, Tc is also highly dependent on environmental conditions, tree density, size and phenological stage [39]. Therefore, the absolute Tc should not be recommended as a plant water stress indicator. What is noteworthy, is this relationship worsened considerably after the DANA event, as high Tc values (in the range of 31 to 37 °C) corresponded to Ψ_stem_ < −1.0 MPa (well-watered trees). Although Ψ_stem_ is considered the best reference indicator of nectarine water status [50], these results revealed a lower sensitivity of this discrete measure in saturated soils produced by the DANA event, which may be also associated with autumn leaf senescence. In addition, the same study showed that in nectarine trees, Tc − Ta registered the highest signal intensity values of the remote sensing indicators studied.

There was a strong dependence between Tc − Ta and Ψ_stem_ (Figure 9). Negative Tc − Ta values obtained by the CTL trees were related to Ψ_stem_ below −1.0 MPa, which corresponded to non-limiting soil water conditions [24,25,26,27,28,29,30]. For a moderate water deficit condition (Ψ_stem_ = −1.5 MPa), the Tc − Ta would correspond to 1.6 °C, while for severe water stress conditions (Ψ_stem_ = −2.5 MPa), the Tc − Ta would be 3.1 °C. Similar results were found by Wang and Gartung [17] in peach trees. After the DANA, R^2^ improved compared to those obtained between Tc vs. Ψ_stem_, confirming that Tc − Ta is less dependent than Tc on the weather conditions. 

Similarly, CWSI successfully tracked the plant water status (Figure 10) with the same goodness of fit as those obtained by Tc − Ta vs. Ψ_stem_. In other studies, when CWSI is compared to Tc − Ta as a proxy for the water status, it can be observed that the CWSI more accurately represents plant water status [44]. This fact can be explained because the NWBS changed because of the DANA event, as only with this type of event a CWSI close to 0 is reached. López-López et al. [51] reported that a robust procedure for determining the CWSI in tree crops is needed to rely on a series of long-term temperature readings (at least two years), given the short-term fluctuations of Tc. Gonzalez-Dugo et al. [16] obtained NWSB from infrared radiometer sensors for three years for the development of the CWSI. In our studied case, the saturated soil conditions together with an adverse meteorological condition caused by the DANA event are not replicable at field conditions.

## 3. Materials and Methods

### 3.1. Experimental Conditions and DANA Event

The experiment was conducted from June to November 2019 (DOY 158–329) in a 0.5 ha orchard of nine-year-old early-maturing nectarine trees (*Prunus persica* L. Batsch) cv. Flariba on GxN–15 rootstock, at the CEBAS–CSIC experimental station in Santomera, Murcia, Spain (38°0603100 N, 1°0201400 W, 110 m altitude). Trees were spaced at 6.5 m × 3.5 m and trained to an open-center canopy. The soil in the 0–0.5 m layer is stony with a clay–loam texture and has an average calcium carbonate content of 45% and organic matter content of 1.4%. The average bulk density was 1.43 g cm^−3^. The volumetric soil water content (θ_v_) at field capacity (FC) and permanent wilting point (WP) were 29% and 14%, respectively, obtained from the textural data [52]. Trees were drip-irrigated with one line per tree row with four pressure-compensated emitters per tree, each delivering 4 L h^−1^, located 0.5 and 1.3 m from the tree trunk. The amount of irrigation water supplied was measured with a pulse flowmeter (Sensus, 121 120 HRI-A, Barcelona, Spain). 

Standard cultural practices (i.e., weed control, phytosanitary treatments, thinning and pruning) were the usual practices for stone fruit trees grown in the studied area. Seasonal fertilizer applications were 83-56-109 UF year^−1^ of N, P_2_O_5_ and K_2_O, respectively, which were applied by a drip irrigation system [53]. The soil was kept free of weeds and was not tilled. Full bloom took place in early February; nectarine fruits were hand-thinned in March and harvested in early May. Nectarine trees were pruned annually during the dormancy period (mid-December). More information on the phenology of early-maturing *Prunus persica* trees is described in Mounzer et al. [54]. More details about the experimental site can be found in Vera et al. [24]. 

On the 12–14 September 2019 (DOY 255–257), an isolated high-altitude depression or DANA event struck the southeast of Spain, including the studied area, very intensely. A total rainfall of 470.4 mm was registered in the CEBAS–CSIC experimental field station in only 24 h. The heavy rainfall caused widespread flooding (Figure 11), which altered the regular course of the experiment, forcing the disconnection of soil moisture capacitance probes installed to avoid short circuits in the system and made it impossible to carry out the measurements of discrete water relations from DOY 255–275. Only infrared radiometer sensors and the weather station were continuously collecting data at that time.

### 3.2. Irrigation Treatments

Two different irrigation treatments were applied: -**Full-irrigated treatment** (**Control, CTL**)**:** during the whole season trees were irrigated based on 100% of the crop evapotranspiration (ET_c_) to ensure non-limiting soil water conditions. The ET_c_ was estimated following the FAO approach by multiplying the crop reference evapotranspiration (ET_0_), using the Penman–Monteith equation [55], by the crop coefficients (K_c_) obtained by [56] at the same location for *Prunus persica* sp. Irrigation was scheduled weekly, and water was applied daily during the night as needed.-**Non-irrigated treatment** (**DRY**)**:** irrigation was suspended from June to November (DOY 158–329), coinciding with the postharvest period of our early-maturing cultivar. Before and after these dates, irrigation rates were the same as for the CTL treatment.

The experimental layout consisted of a completely randomized design with four replications per irrigation treatment, each consisting of six trees (the central four were used for measurements and the others served as guard trees), with a total of 24 trees per irrigation treatment. No active roots were observed more than 1.5 m from the drip line, as revealed by a root distribution study [57]. 

### 3.3. Continuous Measurements

Agro-meteorological data, including the air temperature (Ta), relative humidity (RH), solar radiation and rainfall were recorded following the World Meteorological Organization’s recommendations using an automated weather station located at the CEBAS–CSIC experimental station, less than 100 m from the nectarine orchard (http://www.cebas.csic.es/general_spain/est_meteo.html, accessed on 10 Decemeber 2022) which read values every 5 min and recorded the averages every 15 min. The crop reference evapotranspiration (ET_0_, FAO-56, Penman–Monteith) was calculated hourly [53]. The daily average VPD was calculated from daily maximum Ta and minimum RH data, as described in [21]. The volumetric soil water content (θ_v_) was monitored with multi-depth EnviroScan^®^ capacitance probes (Sentek Sensor Technologies, Stepney, Australia). Four PVC access tubes were installed 10 cm from the emitter located close (50 cm) to the tree trunk in four representative trees, one from each replication (*n* = 4 per irrigation treatment). Each capacitance probe had sensors fitted at 0.1, 0.3, 0.5 and 0.7 m depths and was connected to a radio transmission unit. Values were read every 5 min and averages were recorded every 15 min. The probes were normalized and calibrated for a clay–loam soil [58,59]. The average θ_v_ values in the 0–0.5 m soil profile were calculated, as this depth corresponds to the maximum root density area [60]. Drip gauges were installed below the emitter with the capacitance probe to monitor the real-time irrigation quantities to detect possible mishaps during the irrigation event.

The canopy temperature (Tc) was monitored in real-time using infrared radiometer sensors (model SI—431 series, Apogee Instruments, Inc., Logan, UT, USA). Six sensors were installed (*n* = 3 per irrigation treatment). Each sensor was mounted on a bracket attached to an aluminum pole, installed 3.50 m above the ground surface of the nectarine tree row, facing the middle south-facing side of the tree canopy (Figure 12). The half field of view (FOV) of these sensors was 14°, with a response time of 0.1 s. The sensors were calibrated by the manufacturer, with an accuracy of ±0.2 °C. They read Tc values every 5 min and recorded average values every 15 min.

The following thermal-based indexes were calculated throughout the experiment: (i)Tc − Ta: the difference between the canopy and the surrounding air temperature.(ii)CWSI (Crop Water Stress Index): calculated empirically according to the equation described by Idso et al. [11]:
(1)$$CWSI=\frac{\left(Tc\;\;-Ta\;\;\right)-\left(Tc\;\;-Ta\;\;\right)LL}{\left(Tc\;\;-Ta\;\;\right)UL-\left(Tc\;\;-Ta\;\;\right)LL}$$
where, (Tc − Ta)LL and (Tc − Ta)UL correspond to the lower and upper limit, respectively. The lower limit (LL) was calculated for a given vapor pressure deficit (VPD), from the Non-Water Stressed Baseline (NWSB), which is equivalent to a canopy transpiring at the potential rate (fully irrigated trees with fully open stomata). The upper limit (UL) was obtained by solving the NWSB equation for VPD = 0 and then correcting for the difference in vapor pressure induced by the difference in Tc − Ta (tree under maximum water stress conditions and fully closed stomata). The linear relationship between Tc − Ta and VPD for full-irrigated trees (CTL treatment) was used to establish the NWSB before and after the DANA event occurred. The (Tc − Ta) UL was a fixed value, taken from the maximum Tc − Ta value recorded in stressed nectarine trees during the experiment. Only cloudless days were selected for NWSB determination and CWSI calculations.

(i)Tc’ − Tc: the difference between the canopy temperature of DRY and CTL treatments, respectively.(ii)ΣTc − Ta: the daily integral thermal index for daylight values of Tc − Ta *>* 0.(iii)ΣTc’ − Tc: the daily integral thermal index for daylight values of Tc − ‘Tc *>* 0

Real-time monitoring was possible because all sensors were connected to a radio-transmission unit that sent data to a gateway connected to a cloud-based web server platform (ADCON Telemetry, Vienna, Austria) for data acquisition, processing and visualization.

### 3.4. Discrete Measurements

Plant water status was estimated by measuring the stem water potential (Ψ_stem_) at midday using a pressure chamber (Soil Moisture Equipment Corp. Model 3000, Goleta, CA, USA). Measurements were taken around midday (12:00 and 14:00 GMT + 0) in one healthy mature leaf from each replicate tree of each irrigation treatment (*n* = 4). Leaves were selected from the north face of the tree, near the trunk, and placed in plastic bags covered with aluminum foil for at least 2 h prior to excision, following the recommendations of [22,61]. Measurements were taken every 7–10 days from June to October.

Leaf gas exchange measurements were performed on the same days as the Ψ_stem_, at around 08:00 and 10:00 GMT + 0, on one sun-exposed leaf per replicate and four replicates per irrigation treatment (*n* = 4). The net photosynthesis (P_n_) (μmol m^−2^ s^−1^), stomatal conductance (g_s_, mmol m^−2^ s^−1^) and transpiration rate (E, mmol m^−2^ s^−1^) were measured at the ambient photosynthetic photon flux density (PPFD ≈ 1500 μmol m^−2^ s^−1^) and ambient CO_2_ concentration (Ca ≈ 400 μmol mol^−1^) with a field-portable closed photosynthesis system (LI-COR, LI-6400, Lincoln, NE, USA) equipped with a 6 cm^2^ transparent leaf chamber. From these parameters, the following parameters were obtained: the intrinsic water-use efficiency, as the ratio of P_n_ and g_s_ (μmol mol^−1^) and the instantaneous water-use efficiency, as the ratio of P_n_ and E (µmol mmol^−1^). 

### 3.5. Statistical Analysis

All data were depicted using SigmaPlot v. 14.5 software (Inpixon, PA, USA) and analyzed by one-way ANOVA using SPSS v 9.1 (IBM, Armonk, NY, USA) to discriminate the effect of irrigation. Statistical comparisons were considered significant at *p* < 0.05, as indicated by the Pearson’s correlation coefficient. The degree of agreement of the regressions between variables was assessed by the coefficient of determination (R^2^) and the mean squared error (MSE).

## 4. Conclusions

On-ground measurements with infrared radiometer sensors monitored Tc in nectarine trees during the summer–autumn postharvest. Tc was used to calculate Tc − Ta, Tc’ − Tc, CWSI and the daily integral indexes of Tc − Ta and Tc’ − Tc. The effects of the DANA event interrupted the soil water content recordings, preventing the measurement of Ψ_stem_ and the leaf gas exchange for 20 days. After the DANA event, the soil–plant water status was recovered in the DRY treatment to the levels of well-irrigated nectarine trees. Only the thermal indexes showed significant differences after the DANA, although these were minor differences. In this sense, the thermal indicators were more reliable than the Ψ_stem_ for assessing the plant water stress in saturated soils. 

A progressive increase in the Tc values of the DRY treatment as the soil dried out after the DANA was expected, but it did not happen due to its influence on the evaporative demand and leaf senescence, which required normalizing its absolute value (Tc − Ta). In addition, the new thermal indicator of Tc’ − Tc showed the highest sensitivity for identifying the nectarine water stress. 

The empirical calculation of the CWSI required establishing different NWBS before and after the DANA event due to the increase in the fit of the lower limit (LL), resulting in a higher precision of the CWSI that coincided with a low evaporative demand characteristic of the autumn–winter season. 

This work shows the robustness of canopy infrared thermography to continuously monitor the plant water status under this type of natural weather disaster. The studied thermal sensors were able to provide reliable canopy temperature data in a non-invasive way when other sensors were not working due to DANA-derived effects. 

## Figures and Tables

**Figure 1 plants-12-00907-f001:**
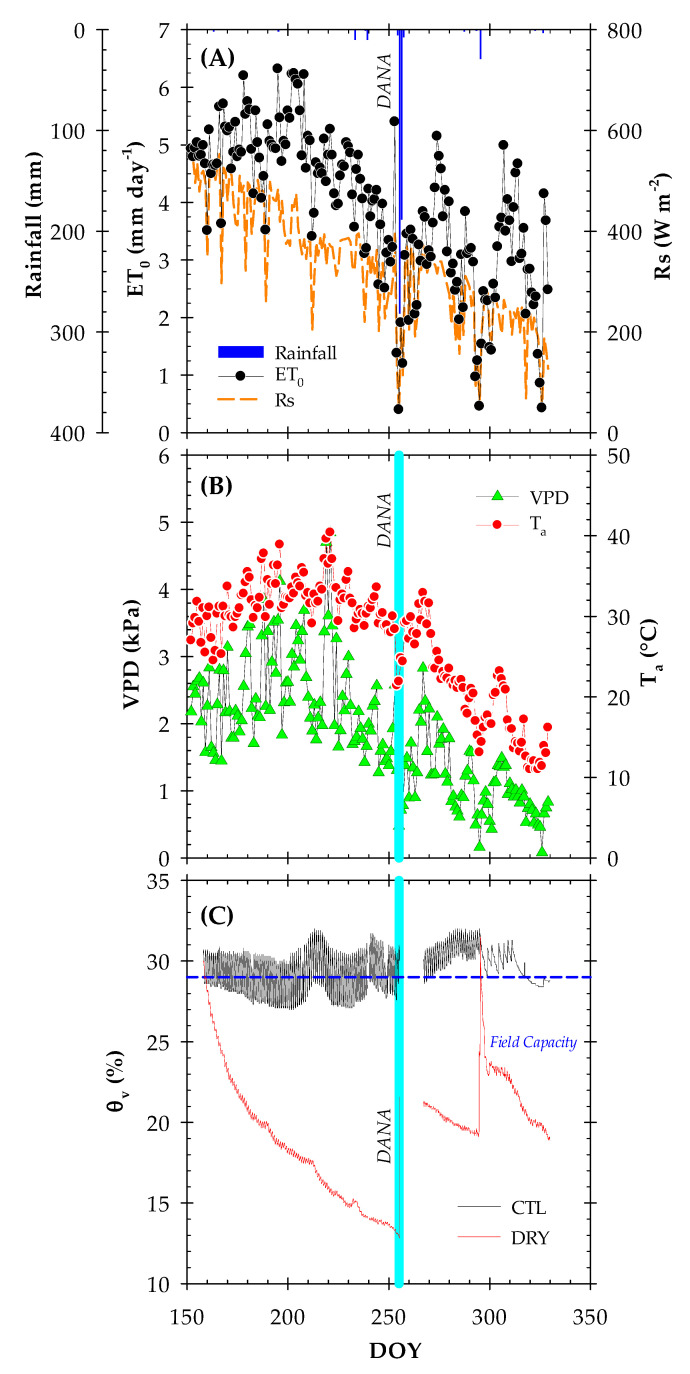
The time course of (**A**) reference crop evapotranspiration (ET_0_), rainfall and solar radiation (Rs), (**B**) vapor pressure deficit (VPD) and daily mean air temperature (Ta) and (**C**) volumetric soil water content (θ_v_) at 0–0.5 m soil depth during the postharvest period (June to November, DOY 158–340) in the CTL (full-irrigated) and DRY (non-irrigated) treatments. Values are the 15-min records corresponding to the average of four capacitance probes. The dashed horizontal blue line delimits the field capacity. The solid vertical cyan line indicates the DANA event.

**Figure 2 plants-12-00907-f002:**
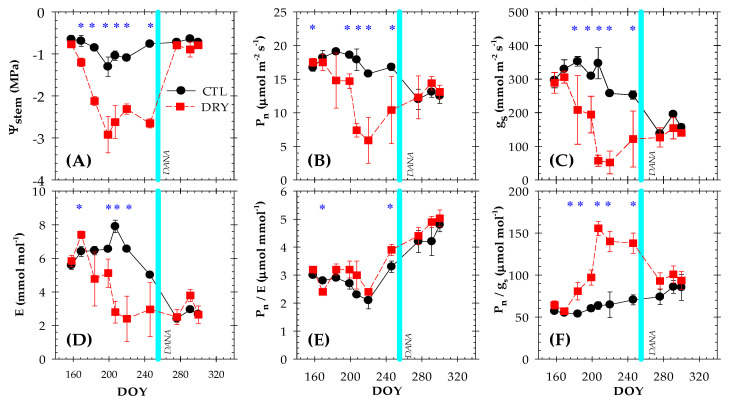
The seasonal variations of: (**A**) midday stem water potential (Ψ_stem_); (**B**) net photosynthesis (P_n_); (**C**) stomatal conductance (g_s_); (**D**) transpiration rate; (**E**) intrinsic water-use efficiency (P_n_/g_s_); and (**F**) instantaneous water-use efficiency (P_n_/E) during the postharvest period (June to November) in the CTL (full-irrigated) and DRY (non-irrigated) treatments. Each point is the mean ± standard error (*n* = 4). The solid vertical cyan line indicates the DANA event. Asterisks indicate statistically significant differences according to LSD_0.05_.

**Figure 3 plants-12-00907-f003:**
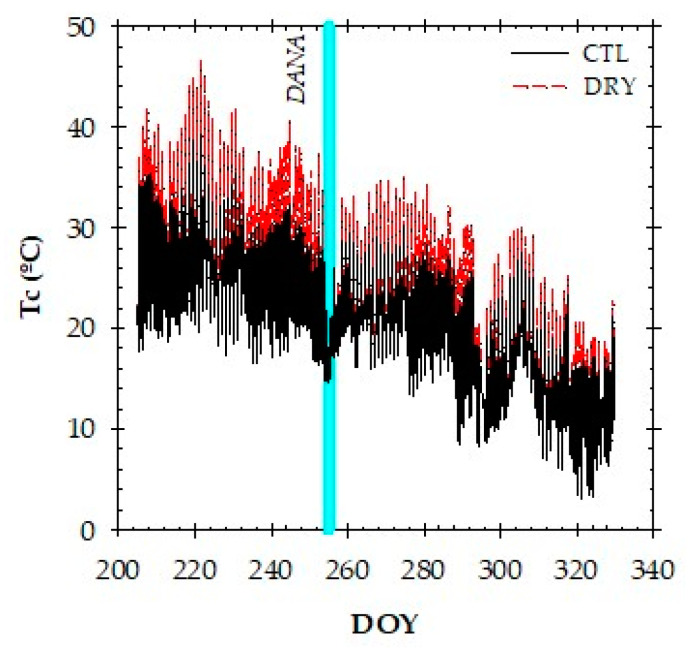
The canopy temperature (Tc) trend in the CTL (full-irrigated) and DRY (non-irrigated) treatments. Values are the average of 15-min recordings of three infrared radiometer sensors. The solid vertical cyan line indicates the DANA event.

**Figure 4 plants-12-00907-f004:**
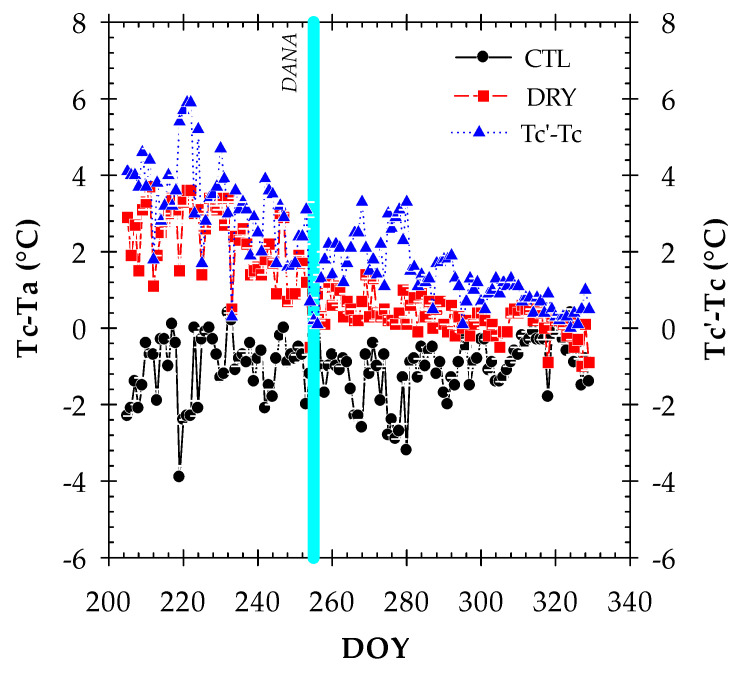
The canopy-to-air temperature difference (Tc − Ta) values in the CTL (full-irrigated) and DRY (non-irrigated) treatments. The canopy temperature difference between DRY (Tc’) to CTL (Tc) treatments. Values are daily means ± SE from three infrared radiometer sensors. The solid vertical cyan line indicates the DANA event.

**Figure 5 plants-12-00907-f005:**
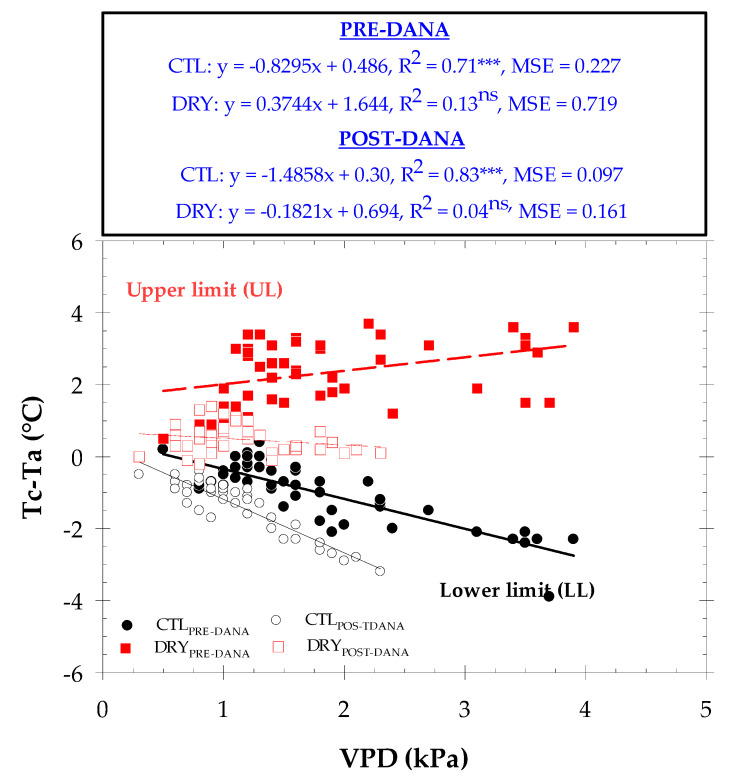
Relationship between Tc − Ta (°C) and measured VPD, averaging from 08:00 to 10:00 GTM + 0. for CTL (full-irrigated) and DRY (non-irrigated) treatments. Each point corresponds to daily cloudless observations including PRE- and POST-DANA periods (*n* = 125). ***: *p* < 0.001, ns: not significant.

**Figure 6 plants-12-00907-f006:**
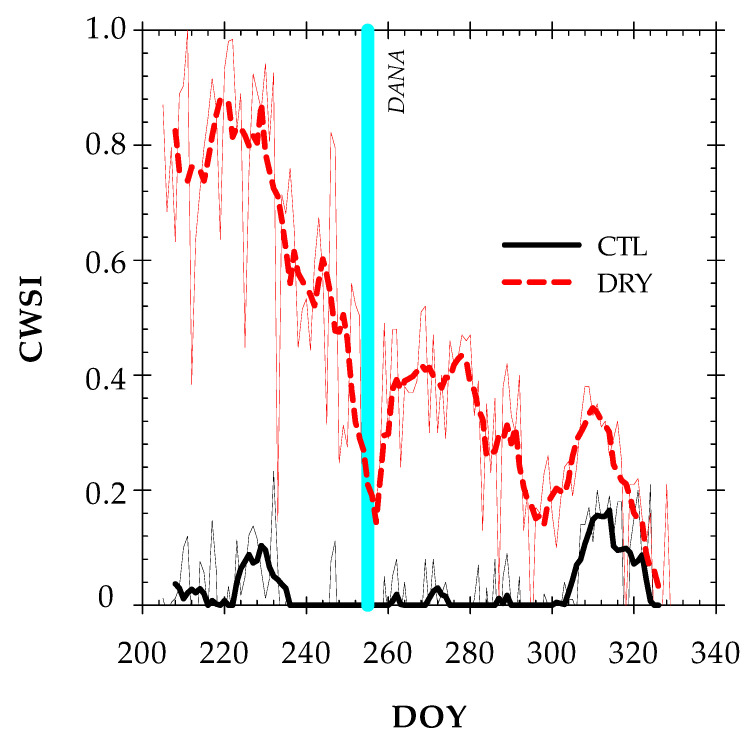
The time course of the Crop Water Stress Index (CWSI) for CTL (full-irrigated) and DRY (non-irrigated) treatments during the experiment. The bold lines show the seven-day moving averages. The solid vertical cyan line indicates the DANA event.

**Figure 7 plants-12-00907-f007:**
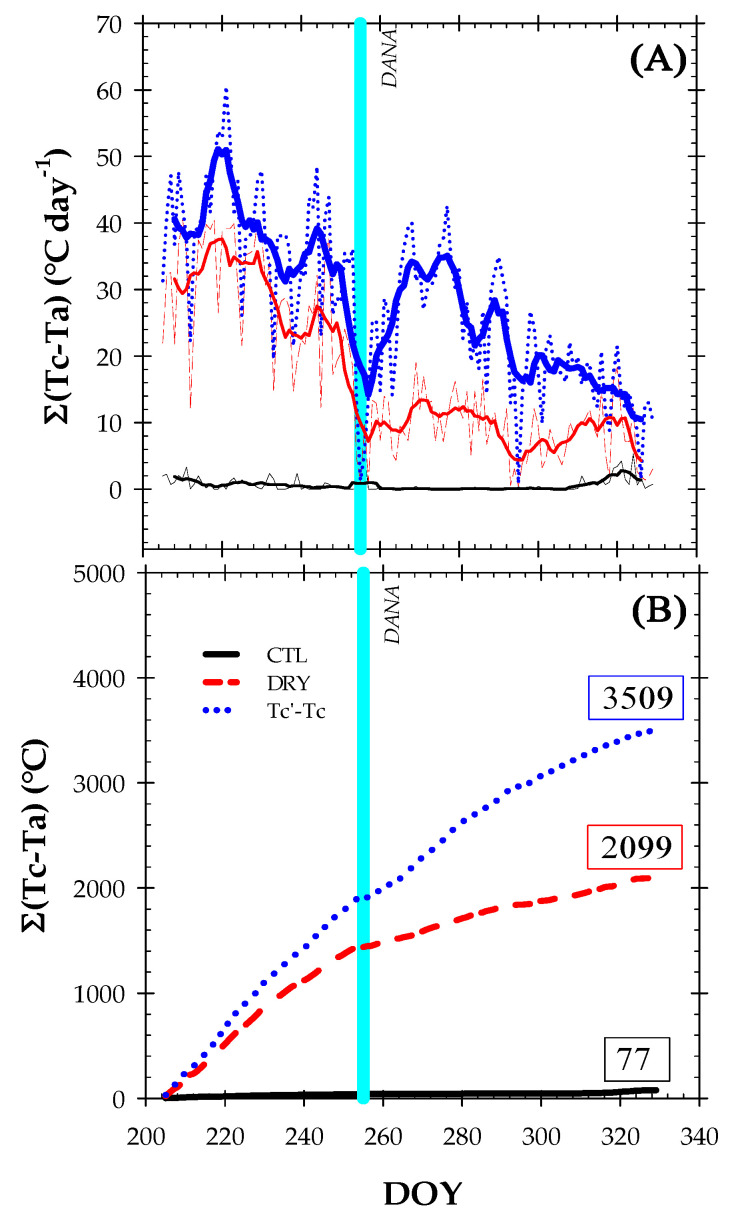
The time course of (**A**) daily ΣTc − Ta and (**B**) integrated daylight ΣTc − Ta during the experiment for the CTL (full-irrigated) and DRY (non-irrigated) treatments, respectively. In (**A**) bold lines indicate the seven-day moving average. The solid vertical cyan line indicates the DANA event.

**Figure 8 plants-12-00907-f008:**
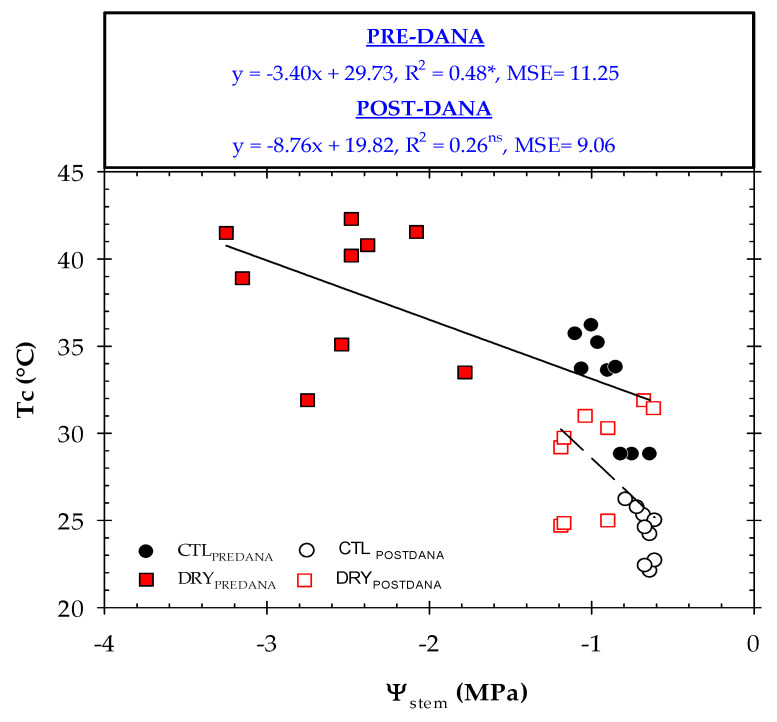
The relationship between Ψ_stem_ and Tc in CTL (circles) and DRY (squares) treatments before and after the DANA, respectively. Tc − Ta values correspond to the average measurements taken between 12:00 and 14:00 GMT + 0. Linear regression and R^2^ are included in the graph. MSE means mean squared error. *: *p* < 0.05, ns: not significant.

**Figure 9 plants-12-00907-f009:**
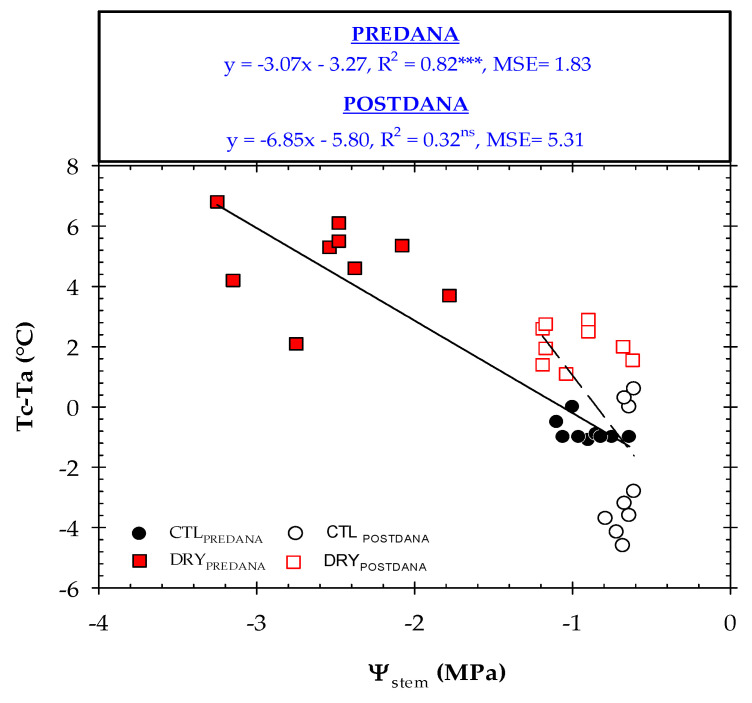
The relationship between Ψ_stem_ and Tc − Ta in the CTL (circles) and DRY (squares) treatments before and after the DANA occurred, respectively. Tc − Ta values corresponded to the average measurements taken between 12:00 and 14:00 GMT + 0. Linear regression and R^2^ are included in the graph. MSE means mean squared error. ***: *p* < 0.001, ns: not significant.

**Figure 10 plants-12-00907-f010:**
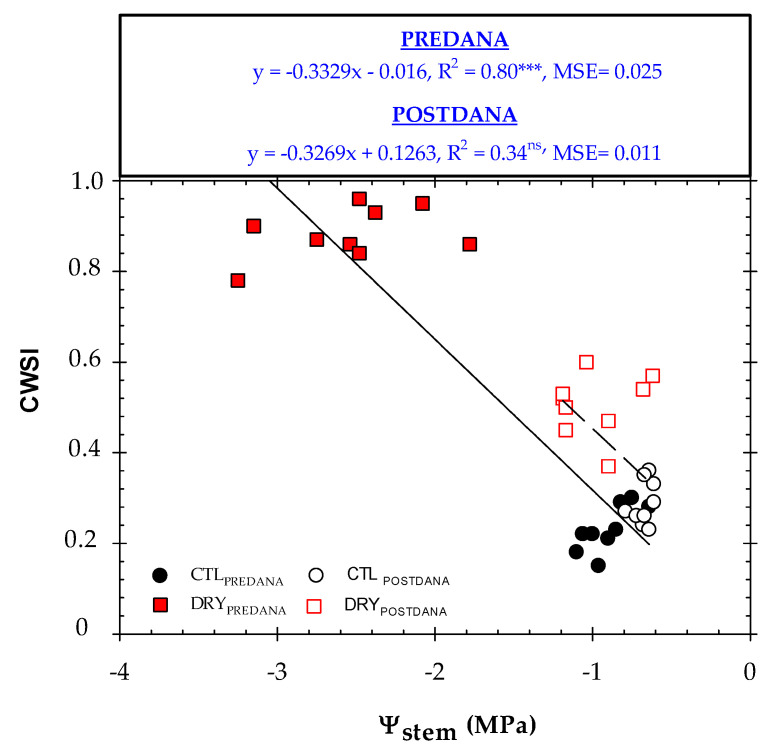
The relationship between Ψ_stem_ and CWSI in the CTL (circles) and DRY (squares) treatments before and after the DANA occurred, respectively. CWSI values corresponded to the average measurements taken between 12:00 and 14:00 GMT + 0. Linear regression and R^2^ are included in the graph. MSE means mean squared error. ***: *p* < 0.001, ns: not significant.

**Figure 11 plants-12-00907-f011:**
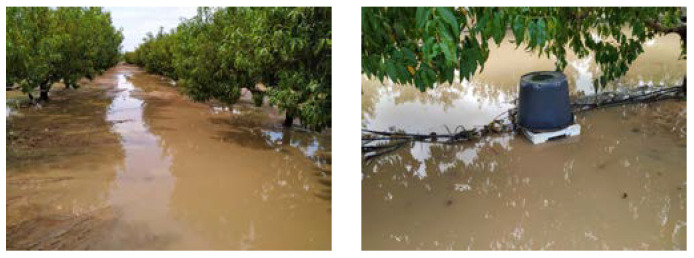
The flooding in the experimental nectarine orchard two days after the DANA event.

**Figure 12 plants-12-00907-f012:**
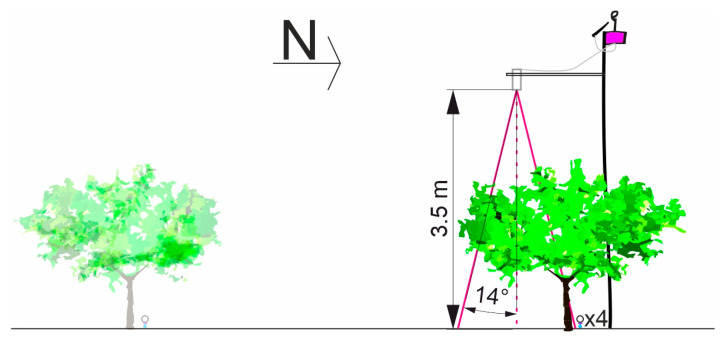
The canopy infrared radiometer sensor placement.

**Table 1 plants-12-00907-t001:** Linear relationships between canopy-to-air temperature difference (Tc − Ta), and the canopy temperature difference between DRY (Tc’) to CTL (Tc) treatments (Tc’ − Tc), vs. the measured VPD at different times of the day before the DANA (PRE-DANA), after the DANA occurred (POST-DANA) and during the whole postharvest experiment (TOTAL) for CTL (full-irrigated) and DRY (non-irrigated) treatments.

	Tc − Ta vs. VPD	TOTAL		PRE-DANA		POST-DANA	
CTL Treatment	TIME (h) (GTM + 0)	R^2^	*p*-Value	R^2^	*p*-Value	R^2^	*p*-Value
	08:00–10:00	0.45	***	0.72	***	0.84	***
	10:00–12:00	0.20	ns	0.19	ns	0.79	***
	12:00–14:00	0.09	ns	0.14	ns	0.82	***
	14:00–16:00	0.00	ns	0.03	ns	0.72	***
	16:00–18:00	0.18	ns	0.07	ns	0.27	ns
	18:00–20:00	0.02	ns	0.01	ns	0.25	ns
	Average	0.01	ns	0.04	ns	0.01	ns
**DRY treatment**							
	08:00–10:00	0.35	***	0.14	ns	0.05	ns
	10:00–12:00	0.43	***	0.20	ns	0.05	ns
	12:00–14:00	0.33	***	0.04	ns	0.10	ns
	14:00–16:00	0.33	***	0.15	ns	0.02	ns
	16:00–18:00	0.41	***	0.40	**	0.11	ns
	18:00–20:00	0.08	ns	0.09	ns	0.36	**
	Average	0.39	***	0.26	*	0.36	**
**DRY–CTL treatments**	**Tc’ − Tc vs. VPD**						
	08:00–10:00	0.80	***	0.77	***	0.85	***
	10:00–12:00	0.74	***	0.56	***	0.84	***
	12:00–14:00	0.56	***	0.15	ns	0.83	***
	14:00–16:00	0.46	***	0.12	ns	0.74	***
	16:00–18:00	0.20	ns	0.26	ns	0.33	*
	18:00–20:00	0.00	ns	0.01	ns	0.61	***
	Average	0.52	***	0.31	**	0.45	***

Values are means of three sensors per treatment. R^2^: coefficient of determination; *, **, *** *p*-value: significant effect at *p* ≤ 0.05, 0.01 and 0.001, respectively; ns = not significant.

## Data Availability

The data presented in this study are available upon request from the corresponding author.

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
