# Peer review of "Impact of a DANA Event on the Thermal Response of Nectarine Trees"

_plants, 2023, doi:10.3390/plants12040907_

Round 1

Reviewer 1 Report

Manuscript: Heavy Rainfall Impact on the Thermal Response of Fruit Trees

 This study addresses the effect of a 3-day- rainfall event (cold water?) on the physiological and thermal response of fieldgrown nectarine trees. The trees were: 1)  well-watered (control) and 2)   subjected to drought conditions. The manuscript is well-written and the results can be of interest to the readers of Plants. I have a few comments and suggestions, which aim to improve the quality of the manuscript. It is not clear in the manuscript how cold were the days  during the DANA event (please see comment on Figure 1). I  suggest considering  that low soil temperature negative affects stomatal conductance and photosynthesis.

 Specifics

 Abstract: In can be improved. I suggest: a rationale 2-3 lines, followed by the Objectives;  Methods; Results and the Conclusion. DANA acronym: I suggest giving it in English.

 Introduction:

= For this reason, different thermal indexes have been performed to minimize the effects of the environmental variables on the absolute Tc values: (i) the difference between canopy and air temperature (TcTa), and (ii) the crop water stress index (CWSI) =

I suggest rephrasing.

 =Subsequently, the empirical approach to calculate the CWSI requires the calculation of nonwater stress and nontranspiration baselines that corresponds to the lower limit (LL) and upper limit (UL) … on clear sunny days and under fullirrigated conditions=

I suggest rewording.

Material and Methods:

= This fact agreed with the stomatal …  values of Pn and gs at earlymorning=

I suggest to use Pn for net photosynthesis instead of

= The net CO2 assimilation rate (ACO2, μmol m−2 s−1)=

 = nearconstant ambient CO2 concentration (Ca ≈ 400 μmol mol−1)=

Instead of "near xx" I suggest providing the standard deviation.

 Results:

Fig.1. It seems that there are too many parameters in Panel A. I suggest considering depicting ToC and VPD in a second panel (Figure). It is a little difficult to see the air temperature data during the DANA event.

 = Shackel et al. [32] observed a Ψstem threshold value of −1.5 MPa, when the decrease in ACO2 was compensated by a reduction in the vegetative apex growth. =

I suggest rephrasing.

= The authors explained this fact due to the decrease in the aspartate amino acid in leaves that affected chloroplasts formation =

I am not sure, if  this explanation necessary.

= The authors concluded that Ψstem was better estimated using remote sensing than from ground measurements, as result of reducing the spatial variability, but … spatial resolution of the proximal ground thermal sensor.=

I suggest rephrasing.

 = Moreover, the study revealed that E, apart from stomatal closure, is also driven by other factors, such as the boundary layer conductance, which is mediated by VPD through changes in wind speed=.

Suggestion: It can be also considered the effect of low soil temperature on gs and  photosynthesis. The paper by Delucia EH (Tree Physiol 2, 143-154,1986)  can be useful.

 If available, I suggest proving soil temperature data

 = After DANA, a progressive increase in the Tc values of the DRY treatment could have been expected as the soil dried, but it does not happen, perhaps due to the progressive decrease in the evaporative demand observed during this period (Figure 1A)=

 I suggest considering the effect of air temperature.

It can be seen in Fig 3:  Drought treatment temperatures are higher the control temperature, both of them declining with time.

= VPD relationship (named NWSB) of CTL nectarine trees.=

Please rewrite. 

Author Response

#REVIEWER 1

Comments and Suggestions for Authors

This study addresses the effect of a 3-day- rainfall event (cold water?) on the physiological and thermal response of field−grown nectarine trees. The trees were: 1) well-watered (control) and 2)   subjected to drought conditions. The manuscript is well-written and the results can be of interest to the readers of Plants. I have a few comments and suggestions, which aim to improve the quality of the manuscript. It is not clear in the manuscript how cold were the days during the DANA event (please see comment on Figure 1). I suggest considering that low soil temperature negative affects stomatal conductance and photosynthesis.

According to these comments, the title has been changed by: Impact of a DANA event on the Thermal Response of Nectarine Trees.  Moreover, the conclusions section has been improved.

 Specifics

Abstract: In can be improved. I suggest: a rationale 2-3 lines, followed by the Objectives; Methods; Results and the Conclusion. DANA acronym: I suggest giving it in English.

Some modifications in the abstract were included. As we explained the meaning of DANA, we prefer using the acronym in Spanish due to the importance of this extreme weather event in our country.

  Introduction:

= For this reason, different thermal indexes have been performed to minimize the effects of the environmental variables on the absolute Tc values: (i) the difference between canopy and air temperature (Tc−Ta), and (ii) the crop water stress index (CWSI) =

I suggest rephrasing.

The text has been rephrasing:

Lines 81-84: For this reason, the canopy−to−air temperature difference (Tc−Ta, Idso et al. [12]), and the crop water stress index (CWSI, Jackson et al. [13]) have been developed to minimize the effects of environmental variables on the absolute Tc values.

 =Subsequently, the empirical approach to calculate the CWSI requires the calculation of non−water stress and non−transpiration baselines that corresponds to the lower limit (LL) and upper limit (UL) … on clear sunny days and under full−irrigated conditions=

I suggest rewording.

The text has been rephrasing:

Lines 93-95: Subsequently, CWSI requires the use of two baselines that relate canopy temperature under maximum stress and non-water stress conditions with vapor pressure deficit (VPD), using data on clear sunny days and full−irrigated conditions [7,9].

Material and Methods:

= This fact agreed with the stomatal …  values of Pn and gs at early−morning=

I suggest to use Pn for net photosynthesis instead of = The net CO2 assimilation rate (ACO2, μmol m−2 s−1)=  = near−constant ambient CO2 concentration (Ca ≈ 400 μmol mol−1)=

Ok. We have changed ACO2 by Pn throughout the text

Instead of "near xx" I suggest providing the standard deviation.

The word near-constant was deleted

 Results:

Fig.1. It seems that there are too many parameters in Panel A. I suggest considering depicting ToC and VPD in a second panel (Figure). It is a little difficult to see the air temperature data during the DANA event.

The Figure has been modified accordingly

 = Shackel et al. [32] observed a Ψstem threshold value of −1.5 MPa, when the decrease in ACO2 was compensated by a reduction in the vegetative apex growth. =

I suggest rephrasing.

OK

Lines 190-192. Shackel et al. [33] observed a Ψstem threshold value of −1.5 MPa associated to a decrease in Pn that was compensated by a reduction in the vegetative apex growth.

= The authors explained this fact due to the decrease in the aspartate amino acid in leaves that affected chloroplasts formation =

I am not sure, if this explanation necessary.

The sentence was deteled

= The authors concluded that Ψstem was better estimated using remote sensing than from ground measurements, as result of reducing the spatial variability, but … spatial resolution of the proximal ground thermal sensor.=

I suggest rephrasing.

OK

Lines 300-305: The authors concluded that Ψstem was a better indicator of plant water status using remote sensing than from ground measurements, because of reducing the spatial variability, but ground measurements better explained the changes in leaf gas exchange than airborne thermal data, due to the higher spatial resolution of the proximal ground thermal sensor.

 = Moreover, the study revealed that E, apart from stomatal closure, is also driven by other factors, such as the boundary layer conductance, which is mediated by VPD through changes in wind speed=.

Suggestion: It can be also considered the effect of low soil temperature on gs and  photosynthesis. The paper by Delucia EH (Tree Physiol 2, 143-154,1986) can be useful.

Thanks for the suggestion. A new paragraph was included in this regard.

Lines 206-209: Although we do not have leaf gas exchange data when DANA event occurred, a decrease in Pn and gs would have been expected due to low root temperatures. According to Delucia [37], the main fact of this reduction is leaf water limitation resulting in an increase in the viscosity of water and in root resistance.

 If available, I suggest proving soil temperature data

We do not have data of soil temperature in the experiment

 = After DANA, a progressive increase in the Tc values of the DRY treatment could have been expected as the soil dried, but it does not happen, perhaps due to the progressive decrease in the evaporative demand observed during this period (Figure 1A) =

 I suggest considering the effect of air temperature.

Done

It can be seen in Fig 3:  Drought treatment temperatures are higher the control temperature, both of them declining with time.

= VPD relationship (named NWSB) of CTL nectarine trees.=

Please rewrite

The sentence is ok. The lower limit is calculated from full-irrigated trees (non-water stress baseline).

The scientific English used in the manuscript has been carefully revised by a native speaker, Mr. Philip Thomas.

Special thanks are due to the reviewer for its constructive remarks

Reviewer 2 Report

The paper is based on a study with high scientific soundness, though the practical utilization of the results is not strongly justified. The manuscript is very much result oriented, the other sections are not in harmony with that part, especially the conclusions. Though the order of sections like in this manuscript is according to the instructions, the classic order of Introduction, Materials and Methods, and Results would make the understanding easier, especially of the abbreviations as some of them are not explained when first mentioned.

 The title of the manuscript is too general, could be specified to the actual study.

The introduction is short referring to the lack of knowledge in the study of thermal response of fruit trees in a DANA event, and also to a hypothesis. Nevertheless, more concrete definition of the goals of the study would be useful.

The figures are complex but self-explanatory.

The results are completed with sufficient discussion, although they are including the study of one DANA event. This arises the question of the generalization or interpretation of the data.

The Conclusion part is very laconic, general, must be improved based on the own results preferably with some practical implications.

Several minor grammar mistakes can be found in the manuscript, I highly recommend its proofreading by a native English speaker.

I inserted several sticky notes in the pdf file of the manuscript with my comments, corrections (not all mistakes), and questions.

Author Response

#REVIEWER 2

The paper is based on a study with high scientific soundness, though the practical utilization of the results is not strongly justified. The manuscript is very much result oriented, the other sections are not in harmony with that part, especially the conclusions. Though the order of sections like in this manuscript is according to the instructions, the classic order of Introduction, Materials and Methods, and Results would make the understanding easier, especially of the abbreviations as some of them are not explained when first mentioned.

 The title of the manuscript is too general, could be specified to the actual study.

The title has been changed by: Impact of a DANA event on the Thermal Response of Nectarine Trees

 The introduction is short referring to the lack of knowledge in the study of thermal response of fruit trees in a DANA event, and also to a hypothesis. Nevertheless, more concrete definition of the goals of the study would be useful.

The information required in the introduction was added. Please see the section. 

 The figures are complex but self-explanatory.

 OK

 The results are completed with sufficient discussion, although they are including the study of one DANA event. This arises the question of the generalization or interpretation of the data.

It is because the saturated soil conditions caused by the DANA event are not duplicable. For this reason, we focus on the effects caused by this type of weather event, and provide the benefits of the thermography in the case of another DANA event occurred. 

The Conclusion part is very laconic, general, must be improved based on the own results preferably with some practical implications.

 The conclusions section was improved accordingly

 Several minor grammar mistakes can be found in the manuscript, I highly recommend its proofreading by a native English speaker.

 The scientific English used in the manuscript has been carefully revised by a native speaker, Mr. Philip Thomas

 I inserted several sticky notes in the pdf file of the manuscript with my comments, corrections (not all mistakes), and questions.

Many thanks. We are going to respond the main comments/questions of the upload pdf here:

 L 33: Justify why "especially" to the Mediterranean basin countries with references.

It has been shown that due to changes in the hydrologic cycle, the incidence of extreme precipitation in a warmer climate can possibly increase even in areas where mean precipitation is projected to decrease. Thus, despite a general decrease in precipitation over most parts of the Mediterranean in the recent past, short and local heavy rainfall events have been recorded in small catchments, many of them near the coast in densely populated areas. This is a common phenomenon in many Mediterranean countries, which are consequently affected by heavy rainfall (DANA) and flash-floods.

The paper by Michaelides et al. (2018) provides an extensive summary of extreme climate events in the Mediterranean, their current status and future projections. It has been included in line 37.

L46: You should involve the statistical probability of such a DANA event occurring in the region as a justification of your study.

New text has been added accordingly

Lines 53-60: A pronounced increase in the percentage of precipitation amount due to heavy (> 30 mm day− 1) and extreme (> 50 mm day− 1) precipitation was also observed during the last decade [1]. Paxian et al. [5] using regional climate models extending up to 2050 showed that in the autumn-winter, there is a distinct tendency towards more intense precipitation extremes in many northern Mediterranean regions, particularly over the Spain and Turkey. Thus, regions with traditional reduced precipitation amounts will be confronted with heavier individual events.

 L90: Though the order of sections is like in this manuscript according to the instructions, the classic order of introduction, materials and methods, results would make the understanding easier.

We have followed the instructions of the journal. But, the abbreviations have been explained when mentioned first time in the R&D section.

Line 93: Specify postharvest

OK

L111-113: As usual in Mediterranean environment, the meteorological conditions during the post−harvest period of early maturing cultivars (May-October).

Line  330: Exatly how? Specify the correlations.

The text has been changed accordingly

Lines 380-382: Given that Tc’−Tc is more sensitive to detect stress conditions than Tc-Ta, our findings showed that Tc’−Tc, can be a suitable plant−based water status indicator for irrigation management in Mediterranean areas with water scarcity.

Line 370: What are the connections with the recent study and what implications can be discussed here?

The text has been rewritten

Lines 436-438: In our studied case, the saturated soil conditions together with an adverse meteorological conditions caused by the DANA event are not replicable at field conditions

L437: 25 cm???

The text has been rewritten

Lines 504-509: Agro−meteorological data, including air temperature (Ta), relative humidity (RH), solar radiation, and rainfall, were recorded following the World Meteorological Organization’s recommendations using an automated weather station located at the CEBAS-CSIC experimental station, less than 100 m from the nectarine orchard (http://www.cebas.csic.es/general_spain/est_meteo.html),

M&M comments to the reviewer

The irrigation applied to the CTL treatment already appeared in line 126 (Results section)

UF: Units of fertilizer

DANA event must be also explained in M&M section

Capacitance probes were disconnected at DANA by the technicians of the CEBAS experimental station.

L 531: Some more implications would be useful here.

The paragraph was completed.

Lines 598-603: The empirical calculation of the CWSI required establishing different NWBS before and after the DANA event due to the increase in the fit of the lower limit (LL), resulting in a higher precision of the CWSI coincided with a low evaporative demand characteristic of the autumn-winter season.

L 535: What novel and practically utilizable results can be gaine from monitoring only?

The last paragraph of the conclusions was completed in this regard

Lines 609-613: This work shows the robustness of canopy infra−red thermography to continuously monitor plant water status under this type of natural weather disaster. The studied thermal sensors were able to provide reliable canopy temperature data in a non-invasive way when other sensors are not working due to DANA-derived effects.

Special thanks are due to the reviewer for its constructive remarks